# Programmable meroterpene synthesis

Xingyu Shen[1,2], Chi P. Ting [1,2], Gong Xu [1] & Thomas J. Maimone [1]*

The bicyclo[3.3.1]nonane architecture is a privileged structural motif found in over 1000 natural products with relevance to neurodegenerative disease, bacterial and parasitic infection, and cancer among others. Despite disparate biosynthetic machinery, alkaloid, terpene, and polyketide-producing organisms have all evolved pathways to incorporate this carbocyclic ring system. Natural products of mixed polyketide/terpenoid origins (meroterpenes) are a particularly rich and important source of biologically active bicyclo[3.3.1]nonane-containing molecules. Herein we detail a fully synthetic strategy toward this broad family of targets based on an abiotic annulation/rearrangement strategy resulting in a 10-step total synthesis of garsubellin A, an enhancer of choline acetyltransferase and member of the large family of polycyclic polyprenylated acylphloroglucinols. This work solidifies a strategy for making multiple, diverse meroterpene chemotypes in a programmable assembly process involving a minimal number of chemical transformations.

---

[1] Department of Chemistry, University of California, Berkeley, 826 Latimer Hall, Berkeley, CA 94702, USA. [2]These authors contributed equally: Xingyu Shen, Chi P. Ting. *email: maimone@berkeley.edu

Nature remains the consummate master assembler of intricate polycyclic molecules, and natural products have —and continue to—provide the impetus for the design and discovery of many of the strategies and tactics pertinent to ring formation in organic synthesis[1]. Embedded in over 1000 natural products is the bicyclo[3.3.1]nonane ring system, which provides a unique spatial arrangement for up to 16 potential substituents to decorate its periphery (Fig. 1a). Despite the significant differences in synthesis capabilities between alkaloid, terpene, and polyketide biosynthetic machinery, all of these natural product classes claim bicyclo[3.3.1]nonane-containing members. Examples include the neuroprotective alkaloid huperzine (1), unusual sesquiterpene upial (2), cytotoxic limonoid mexicanolide (3), and the anti-malarial polyketide rugulosone (4).

Owing to both their sheer number as well as biological importance, polyketide/terpenoid hybrids (meroterpenes) represent the flagship subset of bicyclo[3.3.1]nonane-containing natural products (Fig. 1b). Metabolites formed through the coupling of isoprenoid side chains with either 3,5-dimethylorsellinic acid (DMOA)[2] or acylphloroglucinol derivatives[3,4] constitute the majority of such members. The DMOA-derived members berkeleyone A (5) and berkeleydione (6), which possess anti-inflammatory and caspase-1 inhibitory properties[5,6], and the neuroactive polycyclic polyprenylated acylphloroglucinols (PPAPs) hyperforin (7)[7] and the garsubellins (8–9)[8,9] are representative of the structural complexity these targets possess. Moreover, PPAPs in particular induce wide and diverse biological effects and thus represent lead structures for neuroscience, infectious disease, and oncology drug discovery programs[4]. Hundreds of bicyclo[3.3.1]nonane-containing PPAPs exists and it would appear that nature modulates PPAP function by varying the groups decorating the conserved bicyclo[3.3.1]nonane core. Yet how function is systematically encoded by these groups remains enigmatic. While natural products are unquestionably rich sources of medicinal leads for drug discovery[10,11], in most cases they are still not easily constructed in a step-economical and components-based mix and match format as is typical in many drug discovery programs using small, planar compounds connected through $C(sp^2)–C(sp^2)$ linkages.

Not surprisingly, DMOA-derived meroterpenes[12–14], and especially the PPAPs[15–50], have proven to be popular targets for total synthesis, and a wide range of creative synthetic approaches have been developed for this purpose to date. Herein, we solidify a broad strategy for bicyclo[3.3.1]nonane-containing meroterpene construction, which in the case of PPAPs, offers 10-step entry into synthetically challenging and biologically active type A members in a modular and predictable fashion. Using this strategy, a few simple fragments can be identified as the building blocks for a large proportion of family members. Finally, our work further underscores how following nature's molecular design process can lead to efficient abiotic total synthesis tactics and the development of C–C bond-forming methodology.

## Results

**Synthetic design**. Nature's assembly process of PPAPs served as initial inspiration for our meroterpene synthetic program as it has for others. Although detailed, enzyme-level blueprints are absent for complex PPAPs, labeling studies have provided a likely biosynthetic scenario, which is shown for the garsubellin A nucleus (12) (Fig. 2)[51]. In the initial polyketide assembly phase, three molecules of acetyl-CoA (shown in blue) and one molecule of isobutyryl-CoA (shown in turquoise) are joined and then cyclized to phloroglucinol 10 (Fig. 2a). Notably, the isobutyryl group can be swapped with an assortment of other acyl groups leading to

early diversification of the PPAPs. Next, 10 is merged with three equivalents of dimethylallyl pyrophosphate (shown in green) in a dearomative alkylation event forming polyprenylated acylphloroglucinol 11. Finally, a fourth molecule of dimethylallyl pyrophosphate activates a single prenyl side chain, leading to a presumed cationic intermediate that is trapped by the pendant enol ether event, thus forging two C–C bonds and the hallmark bicyclo[3.3.1]nonane core (see 12) swiftly from 11. To produce garsubellin A (8) from 12, a chemo- and diastereoselective oxidation of a single prenyl side chain can be invoked followed by regioselective opening of the epoxide by the pendant vinylogous acid. Overall, nine C–C bonds are forged in only a few steps.

Our approach toward PPAPs, including 8, did not seek to emulate nature's reaction pathway, but rather took inspiration from the basic building blocks employed (Fig. 2b). In particular, the use of fully intact prenyl groups in the construction of PPAPs had great synthetic appeal both for elegance and simplicity reasons. We envisioned that 2-methylcyclopentenone (shown in blue) could be converted into ketone 13 via straightforward conjugate addition and enolate alkylation chemistry, involving two prenyl equivalents (shown in green) and a one-carbon electrophile (shown in orange). From this simple, but highly substituted, cyclopentanone building block, a reaction was desired that could append on a 4-carbon 1,3-diketone equivalent generating 5,6-fused bicycle 14 in analogy to the Robinson-annulation process, but employing an electrophile of a higher oxidation state. From 14, a formal oxidative isomerization of the 5,6-fused skeleton could construct the bicyclo[3.3.1]nonane skeleton directly (see 14 to 15). Finally, late-stage scaffold-decorating steps, in this case attachment of the final prenyl group and isobutyryl unit, would generate 12. In this approach, seven C–C bonds could be forged in a minimal number of steps.

**Enolate annulation chemistry**. A major hurdle in the implementation of the proposed reaction plan was the four-carbon annulation reaction. Our efforts began with the identification and optimization of a suitable coupling reagent capable of reacting with the enolate of ketone 16 itself, prepared in three steps (Fig. 3a)[43]. Cyclopentanone 16 represents one of the two building blocks that comprise the majority of the PPAPs amenable to our abiotic strategy (vide infra). After significant experimentation, we identified the strained, feedstock chemical diketene (see Supplementary Note 1 for preparation) as a unique and highly reactive four-carbon unit capable of forming bicycle 17 in 11% yield (see Fig. 3a inset, entry 1). Lithium enolates were found to be critical in this process, which also formed C-acylated, but uncyclized, product 18 (3%) as well as O-acylated product 19 (17%). Similar results were also obtained using lithium hexamethyldisilazide (LHMDS) as base (see Fig. 3a inset, entry 2). Given the ability of diketene to react with diisopropylamine, we also evaluated tert-Butyllithium (t-BuLi) and lithium tetramethylpiperidine (LTMP) as bases finding both boosted the amount of annulated product formed (see Fig. 3a inset, entries 3 and 4). In an effort to reduce the O-acylation product, we evaluated ethereal solvents of varying polarity. Although pure diethyl ether showed promise in this regard (see Fig. 3a inset, entry 5), the reaction efficiency was low and almost equal amounts of 17 and 18 were formed highlighting the importance of solvent in the ring-closing step. A variety of mixed ethereal solvents were examined (see Fig. 3a inset, entries 6–8) ultimately arriving at an optimal $THF/Et_2O$ mixture for this specific substrate. After temperature optimization, we were able to produce 17 in 35% isolated yield and as a single diastereomer. It should be stressed that diketene is a highly unique reagent in this annulation process. A variety of other four-carbon conjunctive reagents (see 20–25), including the unstrained diketene

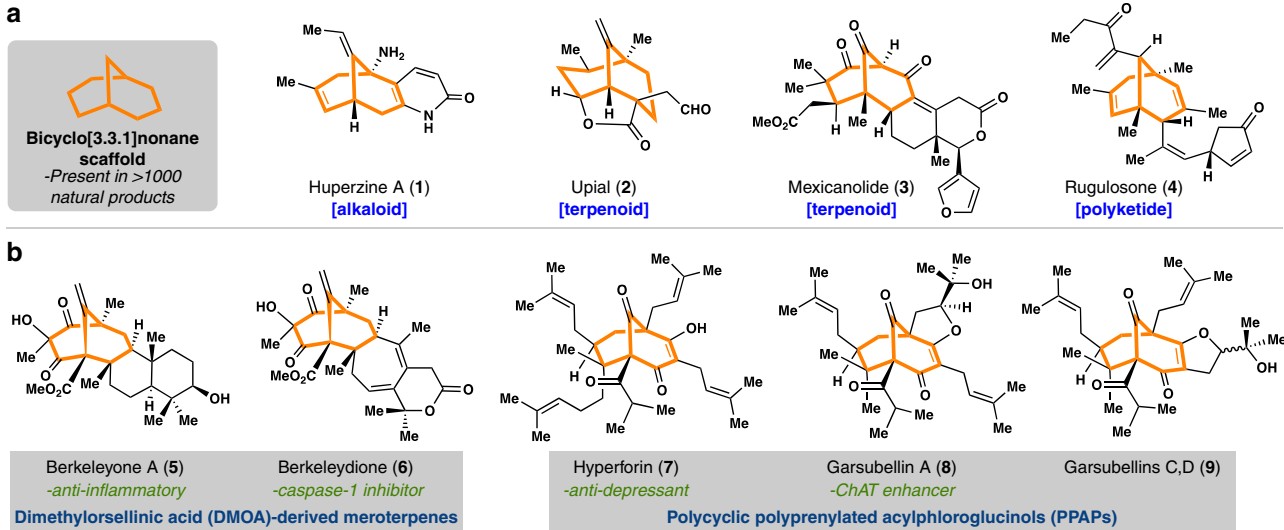

**Fig. 1 Complex bicyclo[3.3.1]nonane-containing natural products. a** Members are found across all areas of natural product space and therapeutic area. **b** The large family of terpene-polyketide hybrids (meroterpenes) as a particularly prolific sources of bicyclo[3.3.1] nonanes.

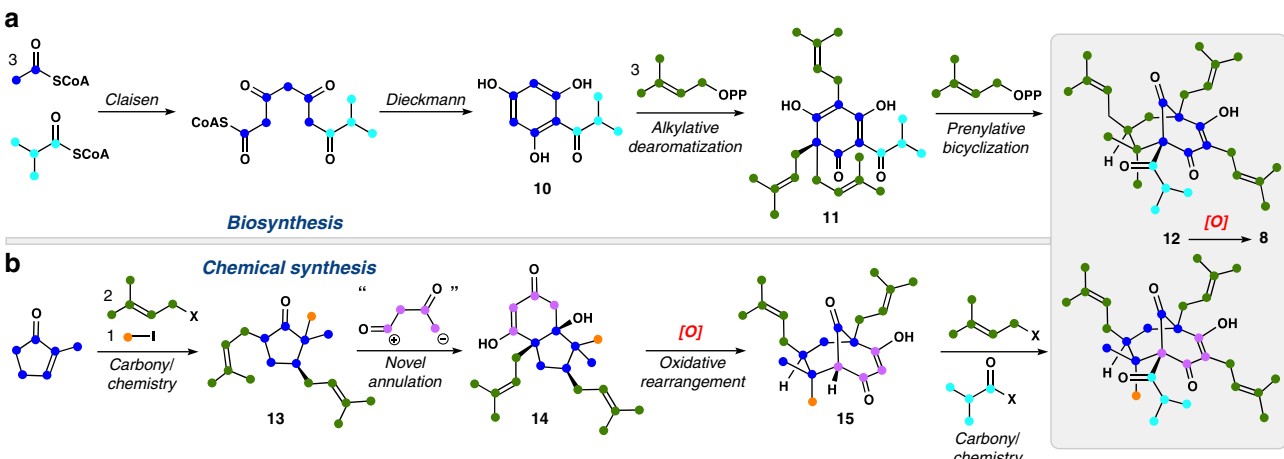

**Fig. 2 Approaches to the synthesis of bicyclo[3.3.1]nonane-containing meroterpenes. a** Nature's presumed synthetic route employing polyketide assembly of an aromatic precursor followed by various C–C bond-forming prenylation events. **b** An abiotic synthetic strategy allows for similarly rapid and modular assembly of complex meroterpene architectures.

acetone adduct (**21**) as well as substituted diketenes (**24**), were all examined and did not produce the desired cyclic product.

The annulation reaction of lithium enolates proved amenable to the preparation of a variety of diverse, cyclic 1,3-diketones (Fig. 3b). The starting enolates can be generated by either reacting the free ketone with strong base (LTMP) or by treating the corresponding silyl enol ether derivative with methyllithium (see Supplementary Notes 3 and 4). Cyclopentanones with varying C-5, C-7, and C-8 substitution patterns (see products **26–29**) were tolerated, highlighting how different inputs can be easily inserted into our PPAP synthetic blueprint. Sterically congested polycycle **30** was also formed in 30% and forms the basis for a DMOA-derived meroterpene synthetic program[13,14]. Diketones **31** and **32**, which contain a *gem*-dimethyl group in close proximity to either an isopropyl or phenyl substituent also highlight the ability of this method to forge highly hindered C–C linkages. A variety of propio- or butyrophenone derivatives with diverse substituents were also annulated, leading to cyclic products **33–42** in good yields (30–65%) and with high diastereoselectivity. We attribute the lower isolated yield of

product **42** to poor lithium enolate solubility in the ethereal solvent mixture employed. Cyclic aliphatic ketones could also be annulated leading to products **43** and **44**; a small preference for the *trans* diastereomer was noted in these cases. Cyclic enone-containing natural products, such as carvone and nootkatone, could be annulated to produce adducts **45** and **46**, respectively. All of the compounds prepared using this chemistry were previously inaccessible chemical entities.

**The total synthesis of garsubellin A.** Although we had previously employed building block **16** in a short synthesis of the antidepressant PPAP hyperforin (**7**)[43], we turned our attention toward garsubellin A (**8**). In a search for naturally occurring small molecules to treat neurodegenerative disease, Fukuyama and co-workers isolated **8** from the wood of *Garcinia subelliptica*, finding that it increased choline acetyltransferase (ChAT) levels 154% relative to control in P10 rat septal neurons[8]. Notably, ChAT is attenuated in neurological pathologies such as Alzheimer's disease[52,53].

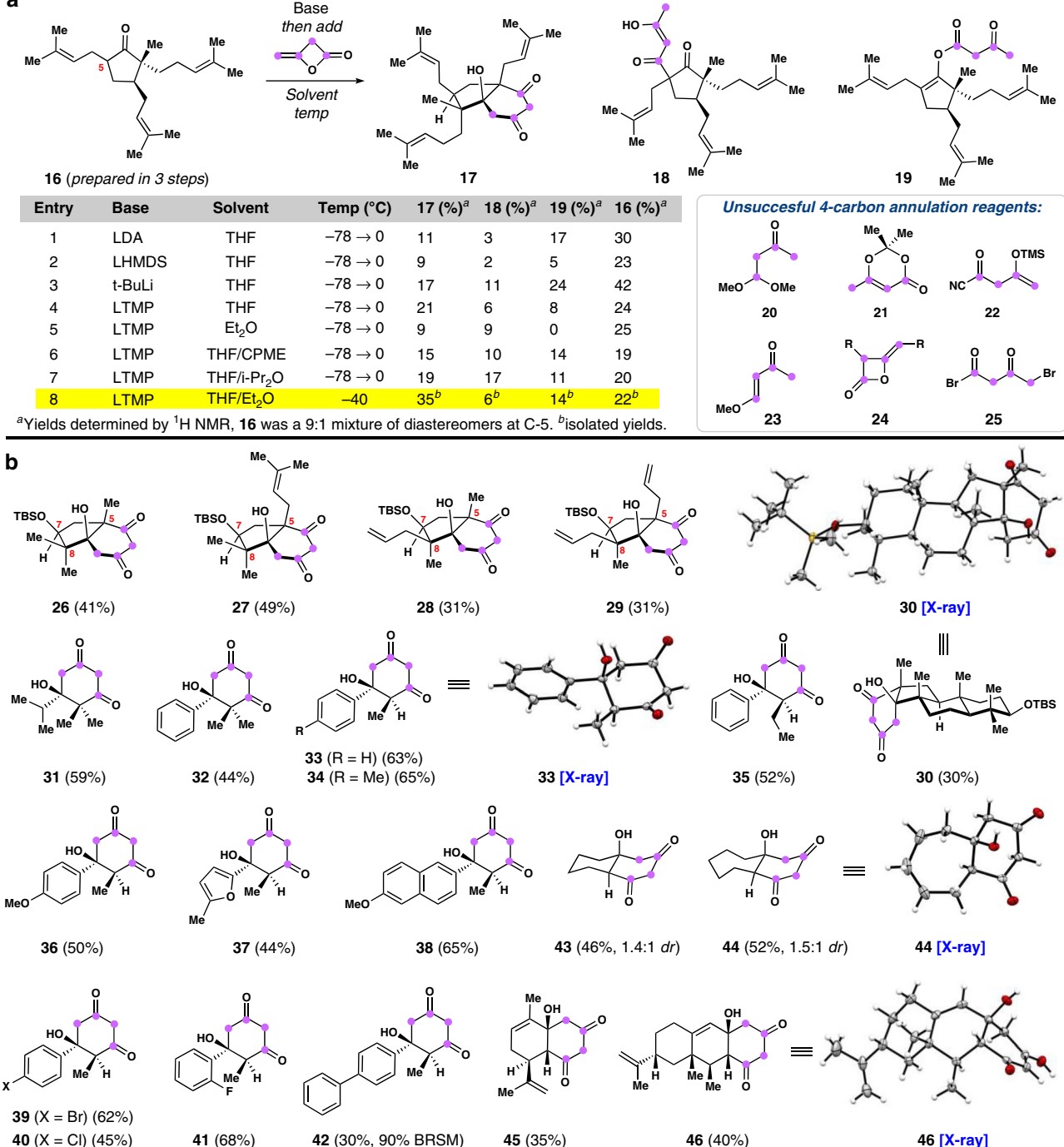

**a**

| Entry | Base | Solvent | Temp (°C) | 17 (%)[a] | 18 (%)[a] | 19 (%)[a] | 16 (%)[a] |
|-------|------|---------|-----------|--------|--------|--------|--------|
| 1 | LDA | THF | −78 → 0 | 11 | 3 | 17 | 30 |
| 2 | LHMDS | THF | −78 → 0 | 9 | 2 | 5 | 23 |
| 3 | t-BuLi | THF | −78 → 0 | 17 | 11 | 24 | 42 |
| 4 | LTMP | THF | −78 → 0 | 21 | 6 | 8 | 24 |
| 5 | LTMP | Et₂O | −78 → 0 | 9 | 9 | 0 | 25 |
| 6 | LTMP | THF/CPME | −78 → 0 | 15 | 10 | 14 | 19 |
| 7 | LTMP | THF/i-Pr₂O | −78 → 0 | 19 | 17 | 11 | 20 |
| 8 | LTMP | THF/Et₂O | −40 | 35[b] | 6[b] | 14[b] | 22[b] |

[a]Yields determined by ¹H NMR, **16** was a 9:1 mixture of diastereomers at C-5. [b]isolated yields.

**b**

Fig. 3 Discovery of a lithium enolate 4-carbon annulation reaction. **a** Key optimization efforts and findings. **b** Successfully prepared cyclohexane-1,3-diones.

Not surprisingly, garsubellin A has attracted worldwide synthetic interest[20,54–59], yet to date, only the groups of Shibasaki[15], Danishefsky[16], and Nakada[36] have documented full total syntheses of this target. Unlike hyperforin, in **8** a single prenyl side chain has been chemo- and stereoselectively oxidized and then cyclized resulting in a complex cyclic ether motif. Mimicking similar oxidative processes in the laboratory in a diastereo-controlled manner has proven challenging in a number of contexts[15,20,36,37], and given the presence of garsubellins C and D in *Garcinia subelliptica* (see **9**, Fig. 1)[9]. Nature may also struggle with this problem in regio- and stereoselective chemical synthesis. Notably, isomers **9** are not reported as ChAT enhancers.

**Synthesis of the key THF substructure.** We began our synthetic efforts by preparing key diprenylated cyclopentanone building block **48** (Fig. 4). In doing so, we developed a straightforward route to this motif, diverging from our previous conjugate addition strategy used to produce **16**. Thus, commercially available 2-methyl-cyclopenten-1-one was first α-alkylated with prenyl bromide and then treated with prenylmagnesium chloride inducing a highly diastereoselective reverse ketone prenylation. The sterically congested tertiary alcohol formed (see **47**) then underwent a facile anionic oxy-Cope rearrangement when exposed to potassium hydride and 18-crown-6. After treatment with MgBr₂, the resulting magnesium enolate was alkylated at carbon with

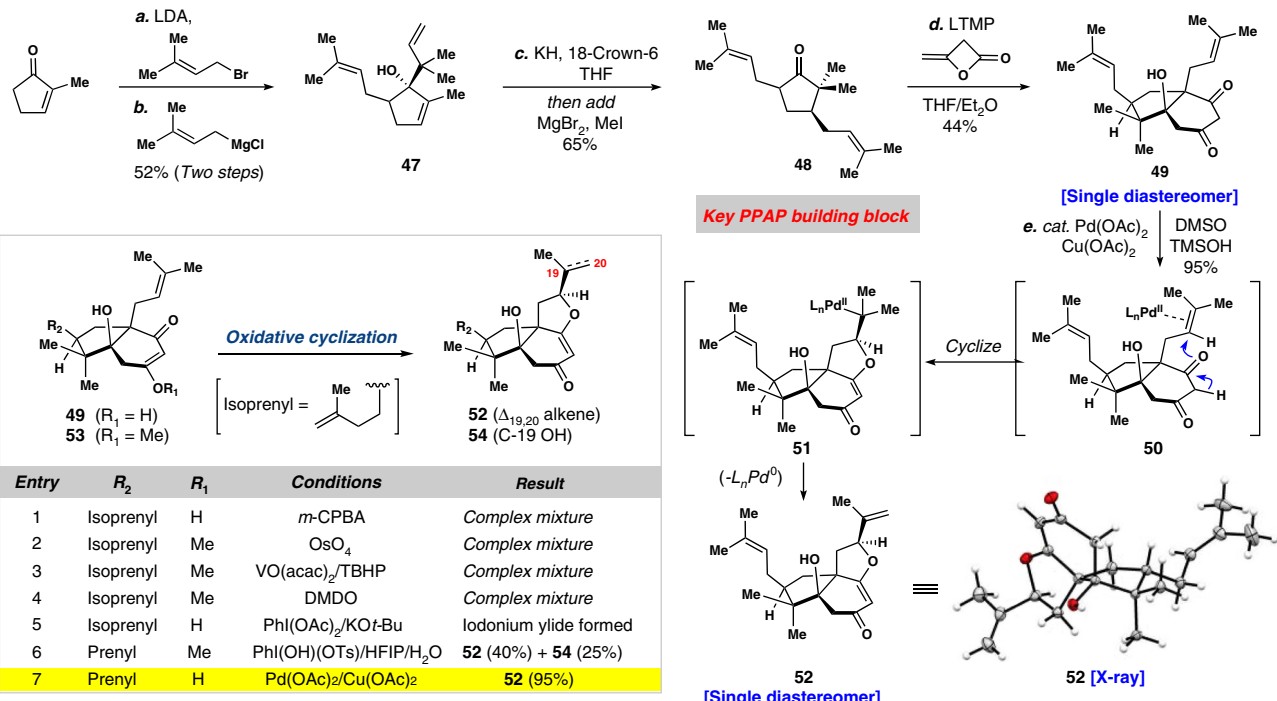

**Fig. 4 Diastereoselective synthesis of a key tricyclic intermediate.** A diketene annulation and unusual vinylogous acid Wacker-type cyclization forge **52** in five chemical steps.

methyl iodide affording **48**. Small amounts (~10%) of the regioisomeric alkylation product as well doubly alkylated materials were also formed in this transformation; whereas these materials could not be cleanly separated by silica gel chromatography, they were effectively removed following the subsequent transformation. To that effect, employing the diketene annulation on the lithium enolate derived from ketone **48** smoothly generated 5,6-fused bicycle **49** in 44% isolated yield and as a single diastereomer; interestingly, this process was noticeably more efficient than with the related ketone **16**.

Although a number of synthetic intermediates could be imagined as suitable starting points for construction of the key tetrahydrofuran (THF) motif embedded in garsubellin A, our final successful route incorporated this structure from bicycle **49** (Fig. 4). After extensive experimentation, we found that catalytic quantities of palladium acetate in the presence of a stoichiometric copper-based oxidant (Cu(OAc)$_2$) led to the formation of **52** as a single diastereomer (confirmed by X-ray crystallographic analysis) in very high yield (95%), presumably via an unusual Wacker-type cyclization of the vinylogous acid (see **50**→**51**→**52**, only an *anti*-oxypalladation pathway is depicted for simplicity)[60]. Typically Pd(II)-activated alkenes are attacked by the carbon atom of 1,3-dicarbonyl-containing compounds[61–64], but geometric constraints presumably disfavor this process in this system.

The construction of this cyclic ether motif has previously proven challenging to access by epoxidation of a prenyl side chain and acid-promoted cyclization—poor diastereoselectivity in the oxidation has historically led to low overall efficiency[15,20,36]. Moreover, facile oxidative degradation of PPAPs further complicates such transformations. We also encountered similar problems in our exploratory studies toward this key ring (see Fig. 4 inset). A variety of oxidation conditions were evaluated on 1,3-diketone-containing substrate **49** or its corresponding vinylogous methylester (**53**). Epoxidation (*m*-CPBA, DMDO, or VO(acac)$_2$/TBHP) and dihydroxylation (OsO$_4$) conditions all afforded complex reaction mixtures containing cyclized and non-cyclized materials; a poorly diastereoselective and chemoselective

oxidation was noted here as well (see Fig. 4 inset, entries 1–4). Seeking to avoid these reaction shortcomings, we reasoned that activation of the prenyl group and direct cyclization by the pendant vinylogous acid in an overall oxidative process would be a more prudent strategy. Interestingly, we initially discovered that treatment of **53** with Koser's reagent (PhI(OH)OTs) in mixtures of hexafluoroisopropanol/water could promote the direct formation of tertiary alcohol **54** (25%) along with alkene **52** (40%) (see Fig. 4 inset, entry 6). Notably, the other electronically similar prenyl group was not oxidized, suggesting this process involved π-bond activation and cyclization; unfortunately, we were unable to further bias the reaction profile toward **54**. Of note, the free 1,3-diketone (i.e., **49**) reacted to form an iodonium ylide under related conditions (see Fig. 4 inset, entry 5). Ultimately, these findings assisted in the discovery of the high-yielding Pd(II)-catalyzed oxidative cyclization of **49**–**52** (see Fig. 4 inset, entry 7).

**Completion of a 10-step synthesis of garsubellin A**. With a robust, five-step route to tricycle **52**, we were poised to evaluate the second key transformation in our programmable route to meroterpenes, namely oxidative isomerization of the 5,6-fused bicycle to the bicyclo[3.3.1]nonane (Fig. 5)[13,43]. Despite the presence of the additional rigidifying THF ring, treatment of this substrate with (diacetoxyiodo)benzene under basic conditions cleanly forged bicyclo[3.3.1]nonane **55** in good yield (75%). Cobalt (II)-salen-catalyzed Mukaiyama hydration of **55** chemoselectively furnished a tertiary alcohol, which could be silylated in the same pot (TMSCl, imidazole, DMAP) to afford **56**[65]. Chlorination of the C-2 position (LTMP, TsCl) allowed for the challenging bridgehead (i.e., C-6) functionalization process (LTMP then *i*-PrCOCl) to proceed smoothly (see **57**→**58**)[66]. We note that in prior work toward hyperforin, only one of two possible vinylogous ester regioisomers took part in the bridgehead deprotonation reaction, an unfortunate finding since they were formed unselectively[43,67]. By incorporating the THF ring prior to the C-6 deprotonation, this problem is obviated in this context[16,18].

**Fig. 5 Total synthesis of garsubellin A.** An oxidative ring expansion constructs the key bicyclo[3.3.1] nonane core framework and enables a 10-step route to the target.

With **58** in hand, all that remained was C-2 prenylation and cleavage of the silyl ether. Transition metal-catalyzed cross-coupling methods offer perhaps the most-attractive solution to this problem owing to their potential for easy and rapid compound diversification late-stage. Yet palladium-catalyzed cross-coupling methods, especially direct prenylation, have historically proven challenging on similar bicyclo[3.3.1] nonane-containing PPAP scaffolds and we assumed deactivated vinyl chloride **58** would be no different[15]. Although recent advances in ligand-controlled, linear-selective prenylation methodology boded well for such endeavors[68–70], the sensitivity of vinylogous ester **58** to hydrolysis and base-induced decomposition proved problematic. After significant experimentation, we successfully applied Buchwald's Pd/CPhos-system[71] to the Suzuki coupling of **58** and prenylBpin, affording garsubellin A (**8**) directly in 57% yield. It should be noted that the successful reactivity window for this transformation was minute and deviating even slightly from the optimized conditions proved quite deleterious. For example, heating the reaction mixture for 8 h instead of four led to an ~10% yield. Similarly, changing the solvent mixture from 1:1 dioxane:H₂O to 4:1 dioxane:H₂O lead primarily to reduction of the vinyl chloride. A small amount of material (17%) still containing the TMS group (see **59**) was also formed under our optimized conditions, but can be converted into **8** via the procedure of Danishefsky[16]. Overall, the synthesis of **8** required 10 steps (3% overall yield) and is nearly half the steps of the previously reported shortest total synthesis of this complex PPAP.

## Discussion

The described total synthesis of garsubellin A combined with our prior work on hyperforin and the DMOA-derived meroterpene berkeleyone A solidifies a general strategy for bicyclo[3.3.1]nonane-containing meroterpene construction and reduces the complexity of many of these targets to that of a substituted cyclopentanone. Recent scholarly analyses of all PPAPs by Grossman and Xu highlighted ~350 natural products with intact bicyclo[3.3.1]nonane ring systems, which can be classified as either type A or type B, depending on the placement of the acyl group (Fig. 6)[3,4]. More than half of these members are further designated "exo-type" PPAPs (see **60** and **61**, Fig. 6) signifying that the C-7 substituent resides on the same face as the C-9

carbonyl group as is found in garsubellin and hyperforin. From the exo-type A grouping, of relevance to the work demonstrated herein, we estimate that only four building blocks, namely **16**, **48**, **62**, and **63** are required to access over 90% of members using the strategy documented here in combination with side chain manipulations. Moreover, building blocks **16** and **48** alone, which we have already successfully employed, cover over 75% of exo-Type A PPAP space. In broader terms, our work in combination with Porco's efficient biomimetic assembly of type B PPAPs[40], and Plietker's modular route to endo-type PPAPs[30,37,38], means that a large swath of all of PPAP chemical space can potentially be covered synthetically in ten or fewer steps.

We note in closing that calls to increase three-dimensional complexity in drug discovery programs are intensifying[72], further highlighting the need for modular routes to spatially multifaceted small molecules in rapid fashion. Nature routinely provides such complexity, and in many cases, also clues to its synthetic design and diversification process. Although directly mimicking such chemistry in the laboratory can prove fruitful, abiotic synthesis strategies also offer opportunities for new reaction discovery and the prospect of accessing diverse naturally occurring secondary metabolites and their derivatives with efficiencies that rival nature.

## Methods

**General**. All air and moisture sensitive reactions were performed in flame-dried glassware under an atmosphere of dry nitrogen or argon. Dry THF, dichloromethane, diethyl ether, toluene, dimethylformamide, and hexane were obtained by passing these previously degassed solvents through activated alumina columns. When mentioned, further solvent degassing was performed by bubbling a stream of argon through the solvent in an ultrasound bath for a period of 10 min. Anhydrous dimethyl sulfoxide was purchased from Aldrich, stored over molecular sieves, and used without further purification. All other reagents were used as received, unless stated otherwise. Reactions were monitored by thin-layer chromatography (TLC) on Silicycle SiliaplateTM glass backed TLC plates (250 μm thickness, 60 Å porosity, F-254 indicator) and visualized by UV irradiation or development with an anisaldehyde or phosphomolybdic/cerium sulfate stain. Volatile solvents were removed under reduced pressure with a rotary evaporator. All flash column chromatography was performed using Silicycle SiliaFlash F60, 230–400 mesh silica gel (40–63 μm). ¹H NMR and ¹³C NMR spectra were recorded with Bruker AV, AVQ, and DRX spectrometers operating at 400, 500, 600, or 700 MHz for ¹H (100, 125, 150, 175 MHz for ¹³C) in CDCl₃, C₆D₆, CD₂Cl₂, CD₃OD, or (CD₃)₂CO. Chemical shifts are reported relative to the residual solvent signal (¹H NMR: δ = 7.26 (CDCl₃); ¹³C NMR: δ = 77.16 (CDCl₃)). NMR data are reported as follows: chemical shift (multiplicity, coupling constants where applicable, number of hydrogens). Splitting is reported with the following symbols: s = singlet, bs = broad singlet, d = doublet, t = triplet, app t = apparent triplet, dd = doublet of doublets, ddd = doublet of doublet of doublets, dt = doublet of triplets, hept = heptet,

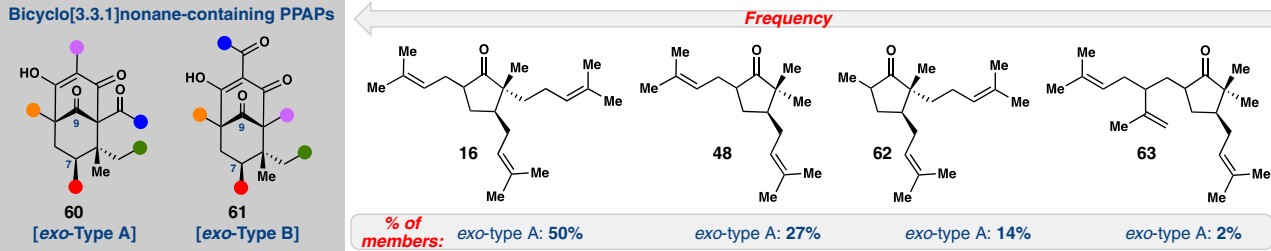

**Fig. 6 Building block analysis of bicyclo[3.3.1]nonane-containing *exo*-type A PPAPs.** Four simple building blocks are embedded in over 90% of members using this strategy.

m = multiplet. IR spectra were recorded on a Nicolet 380 spectrometer as thin films and are reported in frequency of absorption (cm$^{-1}$). High-resolution mass spectra were obtained by the mass spectrometry facility at the University of California, Berkeley using a Finnigan LTQ FT mass spectrometer.

**Experimental data.** For experimental procedures and spectroscopic data of the compounds, see Supplementary Information. For general procedures for preparing diketene and its employment in enolate annulations, see Supplementary Notes 1–4 and Supplementary Fig. 1. For general procedures to prepare garsubellin A and intermediates, see Supplementary Fig. 2 and Supplementary Note 5. For NMR spectra of synthetic intermediates, see Supplementary Figs. 3–70. For the comparison of synthetic and natural garsubellin A NMR spectra see Supplementary Tables 1–2, Supplementary Fig. 71, and the Supplementary References.

## Data availability

The X-ray crystallographic coordinates for structures **30**, **33**, **46**, **44**, and **52** have been deposited at the Cambridge Crystallographic Data Centre (CCDC) with the accession codes CCDC #1521625, 1521624, 1956231, 1521627, and 1879639, respectively. (www.ccdc.cam.ac.uk/data_request/cif) All other relevant data supporting the findings of this study are available within the article and its Supplementary Information files.

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

## Acknowledgements

We thank the National Science Foundation (NSF) for funding this work (CAREER award #155454 to T.J.M.). C.P.T. also acknowledges the NSF for providing a Graduate Research Fellowship (DGE-1106400). G.X. thanks the Swiss National Science Foundation (SNSF) for a postdoctoral mobility fellowship (P2BEP2_162076). T.J.M. is an Arthur C. Cope Scholar and acknowledges unrestricted grant support from Bristol-Myers Squibb, Amgen, Novartis, and Eli Lilly. We thank Dr. Hasan Celik for NMR spectroscopic assistance and NIH grant S10OD024998. Dr. Nicholas Settineri is acknowledged for X-ray crystallographic analysis wherein support from NIH Shared Instrument Grant S10RR027172 is also acknowledged.

## Author contributions

T.J.M. and C.P.T. conceived of the project. X.S., C.P.T. and G.X. performed all of the synthetic experiments. All authors analyzed the data. T.J.M. directed the project and wrote the manuscript with assistance from X.S. and C.P.T.

## Competing interests

The authors declare no competing interests.
