## [Peer Review File · Nature Communications]

Reviewers' comments:

Reviewer #1 (Remarks to the Author):

In this manuscript, Prof. Maimone and co-workers have reported an abiotic annulation strategy for the synthesis of a series of diverse, cyclic 1,3-diketones. This strategy and the followed oxidative rearrangement have been successfully applied into the total synthesis of garsubellin A, an enhancer of choline acetyltransferase and member of the large family of polycyclic polyprenylated acylphloroglucinols. So far, three research groups have completed the total synthesis of garsubellin A, with the synthetic steps range from 17 to 26 steps (Danishefsky, 2005, 17 steps; Shibasaki, 2005, 23 steps; Nakada, 2013, 26 steps). By the current strategy, only 10 steps were required to finish the synthesis. In addition, an innovative Pd(II)-catalyzed unusual Wacker-type cyclization of the vinylogous acid has been developed to construct the key tetrahydrofuran motif in high yield, which has previously proven challenging to access by several groups (ref. 13, 18, 34). Although the group has published the total synthesis of hyperforin (J. Am. Chem. Soc. 2015, 137, 10516) using the similar key strategy, the current effort has demonstrated the application of the method in the synthesis of 27% of the remaining exo-type A PPAPs (Figure 6), which could be an important extension to this area. In order to further improve the manuscript before publication, the following questions should be addressed.

- 1) The superscript "#" on the authors' name is not defined in the manuscript.
- 2) It could be interesting if the author can demonstrate the oxidative rearrangement with the diketone 35 as the product is hard to access by other method.
- 3) A newly published total synthesis that could access to a series of endo-type B PPAPs with the bulky geranyl groups should be cited (Org. Lett. 2019, 21, 8075-8079).
- 4) In the discussion, the author has analyzed the potential synthesis of exo-type A and B PPAPs with current strategy, which concluded that over 90% of the exo-members could be accessed. In the reviewer's opinion, maybe the hypothesis is better to only include the exo-type A as no synthesis or further discussion of the exo-type B was revealed in the manuscript.

Hongxi Xu

Reviewer #2 (Remarks to the Author):

Comments:

Maimone and coworkers have made further contributions the synthesis of bicycle[3.3.1]nonanes and completed the synthesis of garsubellin A, which is substantially shorter than previous syntheses. The work is well written and provides some more detail into the overall strategy and connects with previous work while expanding with new contributions.

Overall I recommend acceptance of this paper with some minor clarification and enhancements.

Here are some questions and comments from a review of the manuscript.

1. (Page 2, Paragraph 2) What structural features make type A PPARs the most synthetically challenging? Perhaps a brief mention could offer some clarification.
2. (Page 3, Paragraph 1) As a general curiosity question, is diketene easy to obtain? Who sells it? Some vendors indicate that this product is discontinued and no longer for sale. This might be of interest to someone who wants to apply the annulation method in their work.
3. (Page 3, Paragraph 1) The more detailed overview of conditions tried and reagents that didn't work is certainly a good story and helpful for synthetic understanding. Figure 3A and 3B expand on the work in Maimone's previous Berkeleyone A paper in JACS. The Table in Figure 3A and unsuccessful reagents constitute valuable new work. However, In Figure 3B, substrates 28, 29, 30,

31, 32, 33, 34, 35, 37, 38, and 39 are previously reported substrates (identical yields and dr) while 26, 27, and 36 appear to be new. If the ratio of THF/Et₂O is important, it worth mentioning. It is stated that lithium enolates are critical in the process. Have the authors tried other enolates such as ones with Na, K, or cations? Additionally, some other ethereal solvents may be worth investigating if interested: MTBE and 2-MeTHF.

4. (Page 3, Paragraph 2) For 1.4:1 dr (substrate 38) and 1.5:1 dr (substrate 39), "predominantly" may be too strong of a term for modest selectivity.

5. (Page 4, Paragraph 2) The differences in reactions needed to elaborate the cyclopentane fragment compared to the hyperforin synthesis are interesting and some good insights are offered here.

6. (Page 4, Paragraph 3) In the Wacker oxidation literature, these reactions can proceed by syn- or anti-oxypalladation. In Figure 4 structure 43, anti-oxypalladation is shown as a possible mechanistic pathway. Palladium doesn't coordinate as well to isolated highly substituted alkenes, so I wonder if a syn-alkoxypalladation pathway could allow the palladium to coordinate to the alkene while being bound to oxygen of the vinylogous acid. Additionally, in Figure 4 conditions e for the unusual Wacker-type cyclization step, TMSOH is shown but its role is unclear. Is it important for the success of the reaction, and if so, what does it do?

7. (Page 5, Paragraph 1) To me, it seems like it is quite a conceptual jump to go from the discussion of entry 6 to entry 7 and from hypervalent iodine to Pd. I wonder if the discussion could be arranged to talk about the eventual successful solution with Pd Wacker-type cyclization last for a smoother and more progressive discussion.

8. (Page 5, Paragraph 3) It is quite surprising that such small changes can lead to such a big difference in reaction outcome.

9. (Page 5, Paragraph 3) The analysis of PPAR building blocks is a useful one and helps put this work, past work, and related work into context. While the two building most prevalent building blocks 16 and 41 have been transformed into bicyclo[3.3.1]nonanes, the most convincing demonstration of the advertised programmable approach to meroterpenes would include elaboration of 53 and 54 to bicyclic structures as well. However, the cited references do show other methods for synthesis of motifs not covered by the reported building blocks.

Formatting Changes Recommended:

1. (Page 1, Paragraph 3) "...garsubellins (8-9)⁸⁻⁹" -- superscript should be a comma for consistency with other formatted references

2. (Page 1, Paragraph 3) "Hundreds of bicycle[3.3.1]nonane-containing PPAPs exists..." -- (exists should be changed to exist for sentence agreement)

3. (Page 2, Paragraph 3) "labelling" -- fun fact learned during review for this paper: labeling is American English spelling while labelling is British English spelling.

4. (Page 2, Paragraph 4) "isobutyl" -- misspelled and should be isobutyryl as in paragraph 3.

5. (Page 3, Paragraph 4) "disease^{51,52}." -- move period before the references.

6. (References) For ref. 6, "et al." is used whereas in other references, all authors are listed. I defer to the editors but I think all authors should be listed here as well. Additionally, correct the spelling of "Meroteprnes" to "Meroterpenes"

7. (References) For ref. 49, please list all authors instead of "et al." for completeness.

8. (References) For ref. 60, there appears to be an extra space before Hegedus.

9. (Figure 3) The figure title has an extra word that should be removed. Perhaps make it "an abiotic synthetic strategy allows..." and remove "employing"

10. (Figure 3A table, Entry 3) "t-Buli" should be "t-BuLi"

11. (Figure 3 and 4) Is there any dr information on compounds 16, 40, and 41? Other readers would be interested for other synthetic applications, even if the bicycle[3.3.1]nonanes are a single diastereomer.

12. (Figure 5) The mechanism for this ring expansion looks different from identical conditions on a similar substrate in Maimone's previous hyperforin synthesis. Perhaps I am missing something, but has the mechanistic understanding changed since then such that a different mechanism is being

proposed?

13. (Experimental Procedures, Supplementary Note 1, Page SI-2). For the chromatography conditions 5% Et₂O in hexanes -> 15% EtOAc in hexanes, should Et₂O be EtOAc?
14. (Experimental Procedures, Supplementary Note 2, Page SI-3). How is LTMP prepared? Usually this is good to include. Or at least cite a reference to indicate which procedure is being followed.
15. (Experimental Procedures, Supplementary Note 2). As a general question, do all of these compounds exist predominantly in the diketone form with little to no enol?
16. (Experimental Procedures, Supplementary Note 2). As a general comment, it seems that melting point is sometimes listed as "m.p." and other times as "mp." IR reporting format is slightly different between sections of the experimental section with "ν_{max}" included in some places but not others. These are relatively minor.
17. (Experimental Procedures, Supplementary Note 2, page SI-7). More major is the missing IR and HRMS for diketone 36. This data should be included for completeness. Additionally, it would be good for other readers to know the dr of enone SI-1, alcohol 40, crude compound 41, if available.
18. (Experimental Procedures, Supplementary Note 2, page SI-11). "MgSO₄" needs subscript. Extra s in "hexaness"
19. (Experimental Procedures, Supplementary Note 2, page SI-13). Remove extra space between "[M+H]⁺ : 453.2222..."
20. (Spectra Section) The spectra do not seem to be formatted consistently. Peak picking on CNMR is definitely helpful and recommended. For HNMR, having integrations is definitely useful for readers, while peak peaking can get cluttered and optional.

Reviewer #3 (Remarks to the Author):

In the present manuscript, Maimone and co-workers report a concise (10 steps) total synthesis of PPAP natural product garsubellin A via a key oxidative isomerization of the 5,6-fused bicycle to the bicyclo[3.3.1]nonane core structure. The tetrahydrofuran moiety was synthesized via an unusual vinylogous acid Wacker-type cyclization. Additionally, a lithium enolate 4-carbon annulation reaction was developed to synthesize the 5,6-fused bicyclic precursor, and a series of 1,3-diketones were prepared. The manuscript and the Supporting Information (experimental details and NMR spectra) were well prepared. I am quite supportive of this elegant natural product synthesis work being published in Nature Communications. The authors should address the following minor comments and concerns below before publication:

1. Additives such as HMPA and DMPU are known to increase the reactivity of lithium enolate. In the annulation reaction (Figure 3), did the authors try to add HMPA or DMPU to optimize the reaction?
2. The oxidative rearrangement of 45 to 46 (Figure 4), the authors proposed an enolate-iodide intermediate, which is different from the previous α-iodide intermediate (J. Am. Chem. Soc. 2015, 137, 10516–10519; Synthlett 2016, 27, 1443–1449). The authors should explain why they speculate two different intermediates.
3. Ref. 9 should be corrected to "Phytochemistry 49, 853–857 (1998)."
4. Ref. 12 should be corrected to "Angew. Chem. Int. Ed. 56, 12498–12502 (2017)."
5. Ref. 43 should be corrected to "Chem. Commun. 51, 2259–2261 (2015)."
6. Ref. 44 should be corrected to "Chem. Eur. J. 21, 3053–3061 (2015)."

Reviewer #1

1) **Comment:** The superscript “#” on the authors’ name is not defined in the manuscript.

Response: This has been added.

2) **Comment:** It could be interesting if the author can demonstrate the oxidative rearrangement with the diketone 35 as the product is hard to access by other method.

Response: This is an excellent suggestion and something we have also considered. To date, however, we have not successfully shifted a non-bicyclic substrate.

3) **Comment:** A newly published total synthesis that could access to a series of endo-type B PPAPs with the bulky geranyl groups should be cited (*Org. Lett.* 2019, 21, 8075-8079).

Response: This work has been added to the citations.

4) **Comment:** In the discussion, the author has analyzed the potential synthesis of exo-type A and B PPAPs with current strategy, which concluded that over 90% of the exo-members could be accessed. In the reviewer’s opinion, maybe the hypothesis is better to only include the exo-type A as no synthesis or further discussion of the exo-type B was revealed in the manuscript.

Response: This is a fair point and we have modified the text to clearly state what we can actually potentially make using the chemistry documented in the manuscript. In addition, we have modified Figure 6 to only show the % of members for the type A skeleton.

Reviewer #2

1) **Comment:** “What structural features make type A PPAPs the most synthetically challenging? Perhaps a brief mention could offer some clarification.”

Response: This was based on several considerations and observations in the field. First, the type A PPAPs have historically required the most synthetic steps to construct—often by a significant margin. For example, hyperforin initially took 51 steps to synthesize and garsubellin A close to 20 steps. Second, biomimetic cyclizations, which typically offer the shortest routes to PPAPs, often do not favor formation of type A skeletons. This is exacerbated by type A PPAPs which possess a quaternary center in close proximity to a bulky isopropyl ketone (such as found in both garsubellin and hyperforin). To the best of our knowledge, these targets have not been made through direct biomimetic cyclization.

We do however, acknowledge that “synthetic challenge” depends on the strategy being used, thus making our original statement more of an opinion than hard fact. To avoid confusion/controversy, we have therefore omitted “the most synthetically challenging” in the text.

2) **Comment:** As a general curiosity question, is diketene easy to obtain? Who sells it? Some vendors indicate that this product is discontinued and no longer for sale. This might be of interest to someone who wants to apply the annulation method in their work.

Response: This is an excellent question. Our initial work used Sigma-Aldrich diketene, but we recognize they have since discontinued this product. Several vendors in China sell it, thus offering an easy solution to researchers there. An *Org. Synth.* Procedure has also been reported to prepare it, but requires quite sophisticated equipment and is not useful for researchers wanting small amounts quickly.

As a service to the synthetic community, we have detailed in the Supplementary information section (new supplementary note 1) a simple, “in-house” protocol to prepare diketene from

inexpensive acetyl chloride requiring only a simple Hickman distillation head for isolation. The text has also been modified to note this preparation is include in the SI.

3) **Comment:** The more detailed overview of conditions tried and reagents that didn't work is certainly a good story and helpful for synthetic understanding. Figure 3A and 3B expand on the work in Maimone's previous Berkeleyone A paper in JACS. The Table in Figure 3A and unsuccessful reagents constitute valuable new work. However, In Figure 3B, substrates 28, 29, 30, 31, 32, 33, 34, 35, 37, 38, and 39 are previously reported substrates (identical yields and dr) while 26, 27, and 36 appear to be new. If the ratio of THF/Et₂O is important, it worth mentioning. It is stated that lithium enolates are critical in the process. Have the authors tried other enolates such as ones with Na, K, or cations? Additionally, some other ethereal solvents may be worth investigating if interested: MTBE and 2-MeTHF.

Response: We have added a significant number of new substrates increasing the substrate scope to >20 examples. Thus, half of the examples of this transformation in the revised manuscript are now new compounds (26, 27, 28, 29, 34, 35, 36, 40, 42, 45, 49).

We have evaluated these other etheral solvents and found no improvement. The solubility of each enolate also dictates to some extent the optimal solvent mixture for the transformation. For certain substrates, pure Et₂O is fine and for others, adding amounts of THF can be beneficial. We have also evaluated Na and K in the form of the bases NaH, NaHMDS, and KHMDS (the K and Na variants of LTMP were not used). These gave messier reactions and we could only detect mainly *O*-acylation in the crude mixture. This is in line with counterion effects on enolate acylation reactions with Mander's reagent as well.

4) **Comment:** For 1.4:1 dr (substrate 38) and 1.5:1 dr (substrate 39), "predominantly" may be too strong of a term for modest selectivity.

Response: We have removed this descriptor.

5) **Comment:** In the Wacker oxidation literature, these reactions can proceed by *syn*- or *anti*-oxypalladation. In Figure 4 structure 43, *anti*-oxypalladation is shown as a possible mechanistic pathway. Palladium doesn't coordinate as well to isolated highly substituted alkenes, so I wonder if a *syn*-alkoxypalladation pathway could allow the palladium to coordinate to the alkene while being bound to oxygen of the vinylogous acid. Additionally, in Figure 4 conditions e for the unusual Wacker-type cyclization step, TMSOH is shown but its role is unclear. Is it important for the success of the reaction, and if so, what does it do?

Response: These are excellent questions. Either could be possible, but both form the same product in this case. The usual mechanistic probes are not valid in this setting since a gem-dimethyl group is formed in the intermediate. We do note that in the Wacker literature, the *syn* oxypalladation is formed in many instances, but these typically involve alcohol nucleophiles. In the case of a phenol (which is closer in pKa to a vinylogous acid as used herein) the *anti* pathway is formed with an LPdCl₂ catalyst and the *syn* pathway formed with a cationic Pd(II) source; Pd(OAc)₂ is likely not cationic under the neutral conditions we used. We have thus added to the text "For simplicity, only an *anti*-oxypalladation pathway is depicted" and have added a reference on precisely this issue.

The use of TMSOH came initially from our attempts to trap the Pd(II) intermediate with a second oxygen nucleophile (TMSOH, H₂O, Allyl alcohol, etc) which could become the garsubellin A tertiary hydroxyl group. Not surprisingly, this never proved possible, but we found that the yield of the alkene was boosted from ~80% yield to 95% yield by applying the TMSOH:DMSO solvent mixture. This could be due to solvation effects since out of the many conditions tried only the TMSOH:DMSO mixture generated a homogenous mixture after the addition of Cu(OAc)₂.

6) **Comment:** To me, it seems like it is quite a conceptual jump to go from the discussion of entry 6 to entry 7 and from hypervalent iodine to Pd. I wonder if the discussion could be arranged to talk about the eventual successful solution with Pd Wacker-type cyclization last for a smoother and more progressive discussion.

Response: These results are discussed somewhat in the order which they occurred in the project. Entry 6 was nice since it formed some of the tertiary alcohol product directly, but only in low yield (25%), a result which we could not boost further. Given that most of the material (40%) was the alkene, this result necessitated a second Mukaiyama hydration step to convert the mixture of products into the tertiary alcohol. Since a second step would be needed, we then focused on just optimizing for the alkene, thus the switch to Palladium which is superior for this type of transformation.

8. **Comment:** It is quite surprising that such small changes can lead to such a big difference in reaction outcome.

Response: We were surprised too. The garsubellin THF ring is quite susceptible to base-mediated hydrolysis. Once the coupling is complete the reaction needs to be stopped. Dioxane is a common, and often optimal, solvent for cross coupling reactions, but can also function as a reducing agent if transmetallation doesn't proceed rapidly.

9. **Comment:** The analysis of PPAR building blocks is a useful one and helps put this work, past work, and related work into context. While the two building most prevalent building blocks 16 and 41 have been transformed into bicyclo[3.3.1]nonanes, the most convincing demonstration of the advertised programmable approach to meroterpenes would include elaboration of 53 and 54 to bicyclic structures as well. However, the cited references do show other methods for synthesis of motifs not covered by the reported building blocks.

Response: As suggested by reviewer #1, we have altered figure 6 and the final discussion to only focus on the type A PPAPs which are the focus of this work. Given this, former substrates 53 and especially 54 (now numbered 62 and 63) form only a very small proportion of type A members (14% and 2% respectively) and we believe of less interest to the readers of this publication. We agree that there are other existing methods to prepare these types of PPAPs. Given we have a route to the products leading to nearly 80% of the relevant members in a minimal number of steps using this chemistry *and* we have previously demonstrated entry into DMOA members, we believe our advertisement has merit.

10) **Comment:** (Page 1, Paragraph 3) "...garsubellins (8–9)^{8–9}" — superscript should be a comma for consistency with other formatted references

Response: This has been corrected

11) **Comment:** (Page 1, Paragraph 3) "Hundreds of bicycle[3.3.1]nonane-containing PPAPs exists..." — (exists should be changed to exist for sentence agreement)

Response: This has been corrected

12) **Comment:** (Page 2, Paragraph 3) "labelling" — fun fact learned during review for this paper: labeling is American English spelling while labelling is British English spelling.

Response: Interesting

13) **Comment:** (Page 2, Paragraph 4) "isobutylr!" — misspelled and should be isobutyryl as in paragraph 3.

Response: This has been corrected

14) **Comment:** (Page 3, Paragraph 4) “disease^{51,52.}” — move period before the references.

Response:

15) **Comment:** (References) For ref. 6, “et al.” is used whereas in other references, all authors are listed. I defer to the editors but I think all authors should be listed here as well. Additionally, correct the spelling of “Meroteprnes” to “Meroterpenes”

Response: We have added these authors and fixed the typo

16) **Comment:** (References) For ref. 49, please list all authors instead of “et al.” for completeness.

Response: We are using Nature house guidelines here since this list has ~75 authors.

17) **Comment:** (References) For ref. 60, there appears to be an extra space before Hegedus.

Response: This has been corrected

18) **Comment:** (Figure 3) The figure title has an extra word that should be removed. Perhaps make it “an abiotic synthetic strategy allows...” and remove “employing”

Response: This has been corrected

19) **Comment:** (Figure 3A table, Entry 3) “t-Buli” should be “t-BuLi”

Response: This has been corrected

20) **Comment:** (Figure 3 and 4) Is there any dr information on compounds 16, 40, and 41? Other readers would be interested for other synthetic applications, even if the bicycle[3.3.1]nonanes are a single diastereomer.

Response: Compound 16 used for screening in Figure 3A was 9:1 dr with the prenyl group being α -configured which has been added to figure 3. Compound 40 was largely a single diastereomer which was added to the text. Compound 41 was also largely a single diastereomer by ¹H NMR, but as noted in the manuscript, it was isolated as an inseparable mixture along with the C-4 methylated isomer and doubly alkylated products making an exact determination difficult.

21) **Comment:** (Figure 5) The mechanism for this ring expansion looks different from identical conditions on a similar substrate in Maimone’s previous hyperforin synthesis. Perhaps I am missing something, but has the mechanistic understanding changed since then such that a different mechanism is being proposed?

Response: These are just differences in whether the hypervalent iodine reagent is drawn O-bound vs. C-bound with the enolate. Given the growing number of reports showing hypervalent iodide favors oxygen, we now tend to favor drawing the O-bound variant. We have no new insight into which is made by spectroscopic or other techniques—both are reasonable intermediates with proper stereoelectronic alignments for fragmentation.

22. **Comment:** (Experimental Procedures, Supplementary Note 1, Page SI-2). For the chromatography conditions 5% Et₂O in hexanes -> 15% EtOAc in hexanes, should Et₂O be EtOAc?

Response: Yes. Correction has been made

23. **Comment:** (Experimental Procedures, Supplementary Note 2, Page SI-3). How is LTMP prepared? Usually this is good to include. Or at least cite a reference to indicate which procedure is being followed.

Response: Supplementary Note 2 did contain the procedure to make LTMP

24. **Comment:** (Experimental Procedures, Supplementary Note 2). As a general question, do all of these compounds exist predominantly in the diketone form with little to no enol?

Response: At least in CDCl₃, nearly all of these compounds are the diketone tautomers. This is also the case in the solid-state structures.

25. **Comment:** (Experimental Procedures, Supplementary Note 2). As a general comment, it seems that melting point is sometimes listed as “m.p.” and other times as “mp.” IR reporting format is slightly different between sections of the experimental section with “vmax” included in some places but not others. These are relatively minor.

Response: we have corrected these to be consistent throughout.

26. **Comment:** (Experimental Procedures, Supplementary Note 2, page SI-7). More major is the missing IR and HRMS for diketone 36. This data should be included for completeness. Additionally, it would be good for other readers to know the dr of enone SI-1, alcohol 40, crude compound 41, if available.

Response: We have added IR and HRMS data as requested for this compound. The dr is not relevant to enone SI-1 since it has only one stereocenter. Compound 40 was largely a single diastereomer which we have now noted in the text. Compound 41 was also largely a single diastereomer by ¹H NMR, but as noted in the manuscript, it was isolated as an inseparable mixture along with the C-4 methylated isomer and doubly alkylated products making an exact determination difficult.

27. **Comment:** (Experimental Procedures, Supplementary Note 2, page SI-11). “MgSO₄” needs subscript. Extra s in “hexaness”

Response: corrected

28. **Comment:** (Experimental Procedures, Supplementary Note 2, page SI-13). Remove extra space between “[M+H]⁺ : 453.2222...”

Response: corrected

Reviewer #3

1) **Comment:** Additives such as HMPA and DMPU are known to increase the reactivity of lithium enolate. In the annulation reaction (Figure 3), did the authors try to add HMPA or DMPU to optimize the reaction?

Response: Yes we did and high *O*-acylation was observed.

2) **Comment:** The oxidative rearrangement of 45 to 46 (Figure 4), the authors proposed an enolate-iodide intermediate, which is different from the previous α -iodide intermediate (J. Am. Chem. Soc. 2015, 137, 10516–10519; Synthlett 2016, 27, 1443–1449). The authors should explain why they speculate two different intermediates.

Response: this question was addressed above.

3) **Comment:** Ref. 9 should be corrected to “Phytochemistry 49, 853–857 (1998).”

Response: corrected

4) **Comment:** Ref. 12 should be corrected to “Angew. Chem. Int. Ed. 56, 12498–12502 (2017).”

Response: corrected

5) **Comment:** Ref. 43 should be corrected to “Chem. Commun. 51, 2259–2261 (2015).”
Response: corrected

6) **Comment:** Ref. 44 should be corrected to “Chem. Eur. J. 21, 3053–3061 (2015).”
Response: corrected

REVIEWERS' COMMENTS:

Reviewer #2 (Remarks to the Author):

I feel points raised in the previous round of review have been satisfactorily addressed and I recommend acceptance of the manuscript.

Reviewer #3 (Remarks to the Author):

The authors have already addressed all of my previous questions and concerns. I enthusiastically support this revised manuscript for publication in Nature Communications.